# Cross-modality Data Augmentation for End-to-End
# Sign Language Translation

**Jinhui Ye**[1]    **Wenxiang Jiao**[4]    **Xing Wang**[4,†]    **Zhaopeng Tu**[4]    **Hui Xiong**[1,2,3,†]

[1]Thrust of Artificial Intelligence, HKUST (Guangzhou), Guangzhou, China
[2]Department of Computer Science and Engineering, HKUST, Hong Kong SAR, China
[3]Guangzhou HKUST Fok Ying Tung Research Institute [4]Tencent AI Lab
jye624@connect.hkust-gz.edu.cn xionghui@ust.hk
{joelwxjiao,brightxwang,zptu}@tencent.com

## Abstract

End-to-end sign language translation (SLT) aims to directly convert sign language videos into spoken language texts without intermediate representations. It has been challenging due to the data scarcity of labeled data and the modality gap between sign videos and texts. To tackle these challenges, we propose a novel Cross-modality Data Augmentation (XmDA) framework to transfer the powerful gloss-to-text translation capabilities to end-to-end sign language translation (i.e., video-to-text). Specifically, XmDA consists of two key components: cross-modality mix-up and cross-modality knowledge distillation. The former one explicitly encourages the alignment between sign video features and gloss embeddings to bridge the modality gap. The latter one utilizes the generation knowledge from gloss-to-text teacher models to guide the spoken language text generation. Experimental results on two widely used SLT datasets, i.e., PHOENIX-2014T and CSL-Daily, demonstrate that the proposed XmDA framework significantly and consistently outperforms the baseline models. Extensive analyses confirm our claim that XmDA enhances end-to-end sign language translation by reducing the representation distance between sign videos and glosses, as well as improving the translation of low-frequency words and long sentences. Codes have been released at https://github.com/Atrewin/SignXmDA

## 1   Introduction

Sign language is an essential communication tool used in deaf communities. Sign language translation (SLT) has made significant progress in recent years (Camgoz et al., 2018; Yin and Read, 2020; Zhou et al., 2021; Chen et al., 2022; Zhang et al., 2023a), with the goal of converting sign language videos into spoken language texts. The conventional approach to SLT uses a cascaded system in which sign language recognition identifies gloss sequences from continuous sign language videos, and gloss-to-text translation (Gloss2Text) converts the sign gloss sequence into written text (Yin and Read, 2020; Camgoz et al., 2020b; Moryossef et al., 2021; Kan et al., 2022; Ye et al., 2023). Nevertheless, it is well-known that the cascaded system has the significant drawback of time delay and error propagation (Chen et al., 2022; Zhang et al., 2023a).

End-to-end SLT models offer an intuitive solution to circumvent the challenges posed by cascaded systems. These models directly translate sign language videos into spoken texts without requiring an intermediary sign gloss representation (Camgoz et al., 2020a; Chen et al., 2022; Zhang et al., 2023a). However, the end-to-end approach faces significant obstacles in practice due to inadequate documentation and resources for sign languages. This shortage of resources results in a high cost of annotated data (NC et al., 2022), which hinders the development of end-to-end models. There have been a few recent attempts to improve the end-to-end SLT performance, including back-translation (Zhou et al., 2021) and pre-training (Chen et al., 2022), to mitigate the issue of data scarcity.

Along this research line, in this work, we propose a novel Cross-modality Data Augmentation (XmDA) approach to improve the end-to-end SLT performance. The main idea of XmDA is to leverage the powerful gloss-to-text translation capabilities (unimode, i.e., text-to-text) to end-to-end sign language translation (cross mode, i.e., video-to-text). Specifically, XmDA integrates two techniques, namely Cross-modality Mix-up and Cross-modality Knowledge Distillation (KD) (§ 2.2). The Cross-modality Mix-up technique combines sign language video features with gloss embeddings extracted from the gloss-to-text teacher model to generate mixed-modal augmented samples (§ 2.3). Concurrently, the Cross-modality KD utilizes diversified spoken language texts generated by the

---

† Corresponding authors: Xing Wang and Hui Xiong.

powerful gloss-to-text teacher models to soften the target labels (§ 2.4), thereby further diversifying and enhancing the augmented samples.

We evaluate the effectiveness of XmDA on two widely used sign language translation datasets: PHOENIX-2014T and CSL-Daily. Experimental results show that XmDA outperforms the compared methods in terms of BLEU, ROUGE, and ChrF scores (§ 3.2). Through comprehensive analyses, we observed two key findings. Firstly, Cross-modality Mix-up successfully bridges the modality gap between sign and spoken languages by improving the alignment between sign video features and gloss embeddings (§ 4.1). Secondly, the Cross-modality KD technique enhances the generation of spoken texts by improving the prediction of low-frequency words and the generation of long sentences (§ 4.2).

The contributions of our work are summarized as follows:

- We propose a novel Cross-modality Data Augmentation (XmDA) approach to address the modality gap between sign and spoken languages and eases the data scarcity problem in end-to-end SLT.

- Comprehensive analysis demonstrates that XmDA is an effective approach to diminish the representation distance between video and text data, enhancing the performance of sign language translation, i.e., video-to-text.

- We evaluate the effectiveness of the proposed XmDA on two widely used SLT datasets and demonstrate that XmDA substantially improves the performance of end-to-end SLT models without additional training data, providing a valuable alternative to data-intensive methods.

## 2 Methodology

We integrate the proposed Cross-modal Data Augmentation (XmDA) technique into the Sign Language Transformers, which is widely used in sign language translation tasks. The proposed framework is illustrated in Figure 1. In this section, we will first revisit the Sign Language Transformers structure (Camgoz et al., 2020b). Then we provide a more comprehensive explanation of the proposed approach, including two essential components: Cross-modality Mixup and Cross-modality

KD. To ensure clarity, we define end-to-end SLT and the notation used throughout this paper.

### 2.1 Task Definition

We formally define the setting of end-to-end SLT. The existing SLT datasets typically consist of sign-gloss-text triples, which can be denoted as $\mathcal{D} = \{(S_i, G_i, T_i)\}_{i=1}^N$. Here, $S_i = \{s_z\}_{z=1}^Z$ represents a sign video comprising $Z$ frames, $G_i = \{g_v\}_{v=1}^V$ is a gloss sequence consisting of $V$ gloss annotations, and $T_i = \{t_u\}_{u=1}^U$ is a spoken language text consisting of $U$ words. In addition, it's worth noting that the gloss $G$ is order-consistent with the sign gestures in $S$, while spoken language text $T$ is non-monotonic to $S$. End-to-end SLT aims to directly translate a given sign video $S$ into a spoken language text $T$.

### 2.2 Sign Language Transformers

Sign Language Transformers utilize the encoder-decoder architecture of transformer networks to recognize and translate sign language $S$ into both sign gloss $G$ and spoken language text $T$ in an end-to-end manner. The original model is composed of four main components: sign embedding, a translation encoder, a translation decoder, and a CTC classifier. In our work, we extend this model by introducing a fifth component, the gloss embedding, which provides a richer representation for the translation process (e.g., gloss-to-text translation).

**Sign Embedding.** In line with recent research (Camgoz et al., 2020b; Zhou et al., 2021), we use pre-trained visual models to extract sign video frame representations. Specifically, we follow Zhang et al. (2023a) to adopt the pre-trained SMKD model (Min et al., 2021) and extract visual representations from video frames. These representations are then projected into the same size as the gloss embedding through a linear layer. It should be noted that the parameters of the pre-trained SMKD model are frozen during the training of the Sign Language Transformers.

**Translation Encoder.** The translation encoder in Sign Language Transformers comprises multi-layer transformer networks. Its input is the embedding sequence of input tokens, such as the representation of sign video frames. Typically, the input embedding sequence is modeled using self-attention and projected into contextual representations $h(\mathcal{S})$. These contextual representations are fed into the decoder to generate the target spoken text translation.

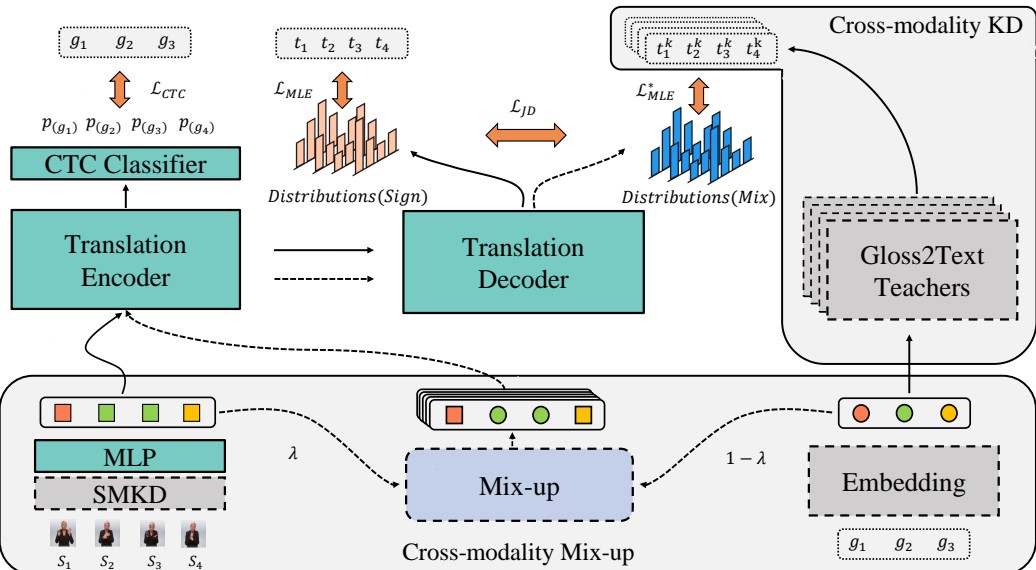

Figure 1: The overall framework of cross-modality data augmentation methods for SLT in this work. Components in gray indicate frozen parameters.

**Translation Decoder.** The translation decoder in Sign Language Transformers involves multi-layer transformer networks that include cross-attention and self-attention layers. These decoder networks produce spoken language sentences based on the sign video representation $h(\mathcal{S})$.

The objective function for spoken language text generation is defined as a cross-entropy loss between the decoder prediction texts and reference texts.

$$\mathcal{L}_{MLE} = -\sum_{u=1}^{|T|} log\, \mathcal{P}\left(t_u|t_{<u}, h(\mathcal{S})\right) \quad (1)$$

**Gloss Embedding.** The gloss embedding, which is presented in Sign Language Transformers, works similarly to traditional word embedding in that it converts discrete gloss tokens into continuous vector representations by an embedding matrix. The gloss embedding is an important expansion to the original Sign Language Transformers (Camgoz et al., 2020b), which brings a broader perspective to the translation process (e.g., gloss-to-text and (sign mix gloss)-to-text).

**CTC Classifier.** Sign Language Transformers inject gloss annotations as the intermediate supervision to train the translation encoder. A linear projection layer followed by a softmax activation (i.e., gloss classifier) is employed to predict sign gloss distribution $\mathcal{P}\left(g_z|h(\mathcal{S})\right)$ of each sign video

frame and model the marginalizing overall possible $S$ to $G^*$ as

$$\mathcal{P}\left(G^*|h(\mathcal{S})\right) = \sum_{\pi \in \Pi} \mathcal{P}\left(\pi|h(\mathcal{S})\right) \quad (2)$$

Where $G^*$ is the ground truth gloss annotations sequence and $\pi$ is a path, i.e., the predicted gloss sequence of sign frames from $s_1$ to $s_Z$, and $\Pi$ is the set of all viable paths resulting to $G^*$ sequence. And the CTC loss function is defined as:

$$\mathcal{L}_{CTC} = 1 - \mathcal{P}\left(G^*|h(\mathcal{S})\right) \quad (3)$$

### 2.3 Cross-modality Mix-up

Cross-modality Mix-up promotes the alignment of sign video features and gloss embeddings to bridge the modality gap by producing mixed-modal representations SLT input. As shown in Figure 1, given a training sample $(\{s_z\}_{z=1}^Z, \{g_v\}_{v=1}^V, \{t_u\}_{u=1}^U) \in \mathcal{D}$, the $\{s_z\}_{z=1}^Z$ and $\{g_v\}_{v=1}^V$ are embedded into a sequence of sign video frames features $\mathcal{F} = [f_1, f_2, \cdots, f_Z]$ and gloss embeddings $\mathcal{E} = [e_1, e_2, \cdots, e_V]$. Note that the dimensions of sign features and gloss embeddings are identical. Cross-modality Mix-up aims to obtain mixed-modal representations $\mathcal{M}$ by combining $\mathcal{F}$ and $\mathcal{E}$.

**CTC Sign-Gloss Aligner.** To perform the Cross-modality Mix-up, an indispensable step is to determine the temporal boundaries between glosses in continuous sign video. Inspired by previous work in speech recognition (Kürzinger et al., 2020),

we employ the CTC classifier as a sign-gloss forced aligner, which returns the start position $l_v$ and end position $r_v$ in the sign video frame flow for each corresponding gloss $g_v$ by maximizing the marginal probability of a specific path $\pi$. More specifically, the sign-gloss forced aligner identifies the optimal path $\pi^*$ from $\Pi$ as follows:

$$\pi^* \Leftarrow \underset{\pi \in \Pi}{\arg\max} \ \mathcal{P}(\pi | h(\mathcal{S}))$$

$$\underset{\pi \in \Pi}{\arg\max} \sum_{v=0}^{V} \sum_{z=l_v}^{r_v} log\mathcal{P}(g_z = g_v^*) \quad (4)$$

Following the best sign-gloss alignment path $\pi^*$, the CTC classifier returns the start position $l_v$ and end position $r_v$ in the sign video frame flow for each gloss $g_v$. With a pre-defined mix-up ratio $\lambda$, we obtain the mixed-modal representations $\mathcal{M}$ as:

$$m_v = \begin{cases} \mathcal{F}[l_v : r_v], & p \leq \lambda \\ \mathcal{E}[v], & p > \lambda \end{cases} \quad (5)$$

$$\mathcal{M} = [m_1, m_2, \cdots, m_V] \quad (6)$$

Where $p$ is sample from the uniform distribution $\mathcal{N}(0, 1)$.

**Mix-up Training.** As shown in Figure 1, to narrow the gap these two types of input representations, namely, mixed-modal input representations $\mathcal{M}$ and unimodal input representations $\mathcal{F}$, we feed them to the translation encoder and decoder, and minimize the Jensen-Shannon Divergence (JSD) between the two prediction distributions. The regularization loss is defined as:

$$\mathcal{L}_{JSD} = \sum_{u=1}^{|T|} JSD\{\mathcal{P}(t_u | t_{<u}, \mathcal{F})$$

$$||\mathcal{P}(t_u | t_{<u}, \mathcal{M})\} \quad (7)$$

### 2.4 Cross-modality Knowledge Distillation

Sequence-level knowledge distillation (Kim and Rush, 2016) (in text mode) encourages the student model to mimic the teacher's actions at the sequence level by using generated target sequences from the teacher. Similarly, Cross-modality KD employs the generative knowledge derived from gloss-to-text teacher model to guide the generation of spoken language text in SLT.

Motivated by the multi-teacher distillation (Nguyen et al., 2020; Liang et al., 2022), we use the data augmentation strategies (Moryossef et al., 2021; Zhang and Duh, 2021; Ye et al., 2023) to train K gloss-to-text teacher models (i.e., $M_{G2T}^1, M_{G2T}^2, \cdots, M_{G2T}^K$) on the given SLT dataset $\mathcal{D} = \{(S_i, G_i, T_i)\}_{i=1}^{N}$. Those teacher models translate each $G_i$ to diversified spoken language translations (i.e., $T_i^1, T_i^2, \cdots, T_i^K$) for the source sign video input $S_i$ and obtain multi-references dataset $\mathcal{D}_{\mathcal{MKD}} = \bigcup_{k=0}^{K} (X_i, G_i, T_i^k)_{i=1}^{N}$. Here, $T_i^0$ represents the original reference and the size of the data is $K + 1$ times that of $\mathcal{D}$.

### 2.5 Overall Framework

**Model Training.** As described in Section 2.4, we first apply Cross-modality KD to enrich the dataset $\mathcal{D}$ into $\mathcal{D}_{\mathcal{MKD}}$ by attaching the diversified spoken language translations $T$. During the training stage, we apply Cross-modality Mix-up (Section 2.3) to produce additional mixed-modal inputs that help bridge the modality gap between the sign video features and gloss embeddings.

As depicted in Figure 1, the end-to-end SLT model is trained by minimizing the joint loss term $\mathcal{L}$, defined as:

$$\mathcal{L} = \mathcal{L}_{MLE} + \mathcal{L}_{CTC} + \mathcal{L}_{JSD} \quad (8)$$

**Model Inference.** In the inference stage, the system utilizes the sign embedding, translation encoder, and translation decoder to directly convert sign video input into spoken language text.

## 3 Experiments

### 3.1 Experimental Setup

**Dataset.** We conduct experiments on two widely-used benchmarks for SLT, namely, PHOENIX-2014T (Camgoz et al., 2018) and CSL-Daily (Zhou et al., 2021). The statistics of the two datasets are listed in Table 9

**Model Settings.** For the baseline end-to-end SLT model, we follow Sign Language Transformers (Camgoz et al., 2020b) consisting of multiple encoder and decoder layers. Each layer comprises eight heads and uses a hidden size of 512 dimensions. For the sign embedding component described in Section 2.2, we adopt the SMKD [1] model (Min et al., 2021) to extract visual features and apply a linear layer to map it to 512 dimensions.

---

[1] https://github.com/ycmin95/VAC_CSLR

| Method | Dev | | | Test | | |
|---|---|---|---|---|---|---|
| | BLEU | ROUGE | ChrF | BLEU | ROUGE | ChrF |
| *Previous Research* | | | | | | |
| Sign Language Transformers (Camgoz et al., 2020b) | 22.38 | - | - | 21.32 | - | - |
| Sign Back-Translation (Zhou et al., 2021) | 24.45 | 50.29 | - | 24.32 | 49.54 | - |
| Contrastive Transformer (Fu et al., 2022) | 21.11 | 47.74 | - | 21.59 | 47.69 | - |
| *Our Experiments* | | | | | | |
| Sign Language Transformers | 22.90 | 48.05 | 44.96 | 22.79 | 47.33 | 44.36 |
| + Cross-modality Mix-up | 23.60 | 48.59 | 46.14 | 23.87 | 48.92 | 46.04 |
| + Cross-modality KD | 24.82 | 50.04 | 47.98 | 24.77 | 49.53 | 48.04 |
| + XmDA | **25.86** | **52.42** | **50.10** | **25.36** | **49.87** | **51.49** |

Table 1: End-to-end SLT performance on PHOENIX-2014T dataset. "+ XmDA" denotes the application of both Cross-modality Mix-up and Cross-modality KD.

| Method | Dev | | | Test | | |
|---|---|---|---|---|---|---|
| | BLEU | ROUGE | ChrF | BLEU | ROUGE | ChrF |
| *Previous Research* | | | | | | |
| Sign Language Transformers (Zhou et al., 2021) | 11.88 | 37.06 | - | 11.79 | 36.74 | - |
| Sign Back-Translation (Zhou et al., 2021) | 20.80 | **49.49** | - | 21.34 | 49.31 | - |
| Contrastive Transformer (Fu et al., 2022) | 14.80 | 41.46 | - | 14.53 | 40.98 | - |
| *Our Experiments* | | | | | | |
| Sign Language Transformers | 11.55 | 36.32 | 11.14 | 11.61 | 36.43 | 11.15 |
| + Cross-modality Mix-up | 15.67 | 42.36 | 14.57 | 15.47 | 42.60 | 14.68 |
| + Cross-modality KD | 17.05 | 44.17 | 15.74 | 16.92 | 43.90 | 15.65 |
| + XmDA | **21.69** | 49.36 | **19.60** | **21.58** | **49.34** | **19.50** |

Table 2: End-to-end SLT performance on CSL-Daily dataset. "+ XmDA" denotes the application of both Cross-modality Mix-up and Cross-modality KD.

In demonstrating the proposed XmDA, we utilize four gloss-to-text teacher models (i.e., $K = 4$), each trained using PGEN (Ye et al., 2023) with varying seeds and subsets of training data. The mix-up ratio for Cross-modality Mix-up is set to 0.6 (i.e., $\lambda = 0.6$).

Furthermore, the gloss embedding matrix is initialized using the most effective gloss-text teacher model as per Table 15, and is fixed constant throughout the training phase. This approach helps to preserve the well-captured distributional properties of gloss representations. Due to space constraints, more details about our model settings and the optimal parameters of each component are detailed in A.2 and A.3.

**Evaluation Metrics.** Following previous studies (Camgoz et al., 2018, 2020b; Zhou et al., 2021;

Chen et al., 2022), we use standard metrics commonly used in machine translation, including tokenized BLEU (Papineni et al., 2002) with 4-grams and Rouge-L F1 (ROUGE) (Lin, 2004) to evaluate the performance of SLT. Following Zhang et al. (2023a), we also report ChrF (Popović, 2015) score to evaluate the translation performance.

### 3.2 Experimental Results

We validate the proposed approaches on the Sign Language Transformers baseline on PHOENIX-2014T and CSL-Daily. The main results are listed in Table 1 and Table 2.

In the PHOENIX-2014T test set, Cross-modality Mix-up surpasses the baseline model by 1.08 BLEU scores, 1.59 ROUGE scores and 1.68 ChrF scores. Similarly, Cross-modality KD yields en-

hancements of 1.98 BLEU scores, 2.20 ROUGE scores and 3.68 ChrF scores. The proposed XmDA method, which integrates both techniques, achieves a gain of up to 2.57 BLEU scores, 2.54 ROUGE scores and 7.13 ChrF scores. The proposed XmDA also outperforms other data augmentation techniques, e.g., Sign Back-Translation and Contrastive Transformer in Table 1, thus demonstrating the effectiveness of the proposed approach.

Table 2 presents the experimental results of the CSL-Daily dataset. A similar phenomenon is observed, indicating the universality of the proposed XmDA.

## 4 Analysis

In this section, we delve into the mechanisms behind the improved end-to-end SLT performance achieved by the XmDA method. We investigate our approach's impact on input space and output translations, focusing on the effectiveness of the Cross-modality Mix-up and Cross-modality KD techniques and their influence on word frequency and sentence length.

### 4.1 Impact of Cross-modality Mix-up

In Section 2.3, we claim the XmDA approach can bridge the modality gap between video and text and promote the alignment of sign video features and gloss embeddings. To investigate this, we conduct experiments on the SLT model using XmDA in Table 1.

We visualize multi-view representation (sign, gloss, mixed-modal) for each PHOENIX-2014T test set sample. We compute sentence-level representations for each input view by averaging embedding sequences and apply t-SNE (Van der Maaten and Hinton, 2008) for dimensionality reduction to 2D. We then plot the bivariate kernel density estimation based on 2D sequence-level representations. As shown in Figure 2, the 2D distributions of sign video features (blue) and gloss embeddings (red) are distinct, while mixed-modal representations (green) lie between them, indicating that mixed-modal representations serve as an effective bridge between sign and gloss representations.

To further investigate the benefits of mixed-modal sequence input, we examine the topological structure of input embeddings for the baseline, "+Cross-modality Mix-up", and the best gloss-to-text teacher model. We employ t-SNE to visualize those embedding sequences in 2D. As de-

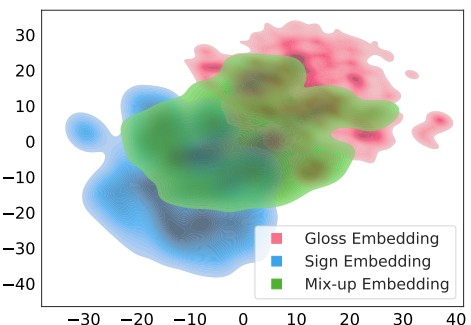

Figure 2: Bivariate kernel density estimation visualization of sentence-level representations: sign embeddings from baseline SLT, gloss embeddings from the gloss-to-text teacher model, and mixed-modal representations obtained by mixing sign embeddings and gloss embeddings with $\lambda = 0.6$. Best viewed in color.

picted in Figure 3, gloss embeddings in the gloss-to-text teacher model have a discrete distribution (in red), indicating excellent representation for different glosses. In contrast, the sign embedding of the baseline SLT model exhibits a smoother distribution (in blue), potentially causing confusion when translating sign language videos into spoken texts. Remarkably, the sign embeddings extracted from the "+Cross-modality Mix-up" model (in green) exhibit a distribution similar to that of gloss embeddings.

For quantitative analysis, we employ Kernel Density Estimation (KDE) to estimate the probability density functions for these three types of embeddings (Moon and Hero, 2014). The resulting entropies from these KDEs are tabulated below:

| Embedding Type | KDEs Entropy |
|---|---|
| Gloss Embeddings | 0.19 |
| Sign Embeddings (Baseline) | 2.18 |
| Sign Embeddings (XmDA) | 0.84 |

Table 3: Entropies resulting from KDEs on different types of embeddings.

Additionally, to provide a quantitative measure of alignment, we compared the average Euclidean distance and cosine similarity at the word-level between sign embeddings and gloss embeddings. The results are presented in the table below:

This demonstrates that incorporating the mixed-modal inputs in the training process enables better sign representation by utilizing the constraint from the teacher gloss embeddings.

| Comparison | ED | Cos Sim |
|---|---|---|
| Sign (Baseline) vs. Gloss | 14.6 | 0.19 |
| Sign (XmDA) vs. Gloss | 8.68 | 0.34 |

Table 4: Average Euclidean distance (ED) and cosine similarity (Cos Sim) between sign embeddings and gloss embeddings.

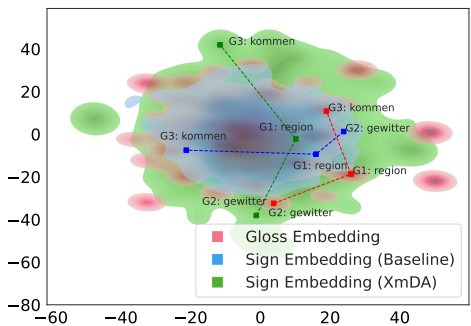

Figure 3: Visualization of gloss and sign representation distributions for the Baseline SLT (in blue) and "+ Cross-modality Mix-up" (in green) models by t-SNE. Best viewed in color.

To illustrate this point, we sample a gloss annotation sequence (e.g., {"$region$", "$gewitter$", "$kommen$"}) from the test set and plot the relative sign frame embeddings and gloss embeddings in Figure 3. Notes, middle frames are selected as representative embeddings based on sign-gloss forced aligner temporal boundaries (§ 2.3). As seen, the Cross-modality Mix-up technique (green dot line) significantly improves the sign embedding representations over the baseline (blue dot line), bringing sign embedding representations within visual and text modalities closer and a better alignment.

## 4.2 Impact on SLT Translation Outputs

To gain a deeper understanding of how the proposed approach enhances the end-to-end SLT, we analyze the translation outputs of PHOENIX-2014T, as shown in Table 1, using the compare-mt toolkit[2]. We focus on two aspects: word frequency and sentence length.

**Words Frequency:** Multiple reference translations have been shown to improve translation quality by promoting diversity and robustness in translation outputs (Nguyen et al., 2020). We wonder how such diversity and robustness benefit

[2] https://github.com/neulab/compare-mt

| Data | Word Frequency | | |
|---|---|---|---|
| | Low | Medium | High |
| Baseline SLT | 30.70 | 52.24 | 60.61 |
| + Xm Mix-up | 31.79 | 55.20 | 60.09 |
| + Xm KD | 33.36 | 55.96 | 61.60 |
| + XmDA | 34.68 | 56.07 | 61.93 |

Table 5: Prediction accuracy (F1 score) of target words in the test set with respect to word frequency. Xm indicates Cross-modality. As the higher F1 score, the better, we mark the improvement by green and degradation by red background.

the prediction of low-frequency words. Specifically, we first categorize the vocabulary into three groups based on the word frequency in the training data, including High: frequency $\in [2000, +\infty)$; Medium: frequency $\in [100, 2000)$; Low: frequency $\in (0, 100]$. Then, we utilize compare-mt to calculate the prediction accuracy of target words, e.g., F1 score (Snover et al., 2006), in the test set with respect to the three groups.

Table 5 presents the results for the different approaches concerning word frequency. It can be observed that the Cross-modality Mix-up and Cross-modality KD methods independently improve the F1 score for low-frequency and medium-frequency words compared to the Baseline SLT. When combined, XmDA method, which stacks both Cross-modality Mix-up and Cross-modality KD, achieves the most significant improvement, particularly in low-frequency words, demonstrating its effectiveness in predicting these words. For high-frequency words, both Cross-modality KD and XmDA improve F1 score, whereas Cross-modality Mix-up shows a slight decrease. These results highlight the potential of cross-modality techniques in enhancing the model's ability to predict low-frequency words, which contributes to a more robust and diverse translation output.

**Sentence Length:** We investigate the translation quality of examples with varied lengths, which can be biased when generating target spoken language texts. Similar to word frequency, we also use the compare-mt toolkit to categorize the examples of the test set into three groups based on the sentence length, including Long: $(20, +\infty)$ tokens; Medium: $(10, 20]$ tokens; Short: $(0, 10]$ tokens.

Table 6 presents the results concerning sentence length. It is well-known that longer sentences

| Data | Sentence Length | | |
|---|---|---|---|
| | Short | Medium | Long |
| Baseline SLT | 36.70 | 17.56 | 12.73 |
| + Xm Mix-up | 38.96 | 18.78 | 13.68 |
| + Xm KD | 34.32 | 19.67 | 15.23 |
| + XmDA | 38.05 | 20.80 | 17.54 |

Table 6: Translation quality (BLEU score) of examples in the test set with respect to sentence length.

are more challenging to translate (Zheng et al., 2020). Cross-modality Mix-up and Cross-modality KD methods stand out, exhibiting significant improvement in translation quality for medium and long sentences. However, the Cross-modality KD method noticeably degrades performance on short sentences. In contrast, the XmDA method significantly improves translation quality for medium and long sentences without compromising the performance of short sentences. This demonstrates that the XmDA approach, which combines the strengths of both Cross-modality Mix-up and Cross-modality KD, offers superior stability across various sentence lengths and effectively addresses the distribution of sentence length in translation tasks better than using the individual methods alone.

### 4.3 Case Examination of Model Outputs

To gain insights into our model's performance in specific scenarios, we present two notable observations. Firstly, our model exhibits strong proficiency in generating low-frequency words, which are often challenging for many translation systems (Table 7). Additionally, our approach showcases remarkable capability in translating long sentences (Table 8), which is another common hurdle in translation tasks.

## 5 Related Work

### 5.1 Sign Language Translation

Existing approaches to Sign Language Translation (SLT) can be categorized into two main types: cascaded and end-to-end methods (Camgoz et al., 2020b). Cascaded methods decompose SLT into two separate tasks (Yin and Read, 2020): sign language recognition, which recognizes the gloss sequences from continuous sign videos, and sign language gloss translation, which translates the gloss sequences into spoken language text.

Conversely, End-to-end SLT methods convert sign videos directly to natural text without using gloss as an intermediary representation. Existing works attempt to formulate this task as a neural machine translation (NMT) problem (Camgoz et al., 2018, 2020a; Zhou et al., 2021; Chen et al., 2022; Zhang et al., 2023a). However, unlike NMT, which benefits from a large-scale parallel corpus (Wang et al., 2022), end-to-end SLT greatly suffers from data scarcity.

Recent studies have focused on the challenge of data scarcity, e.g., sign back-translation (Zhou et al., 2021), transfer learning (Chen et al., 2022) and multi-task learning (Zhang et al., 2023a). Along this research line, this work proposes a novel Cross-modality Data Augmentation (XmDA) framework, which is a more valuable and resource-efficient alternative data augmentation method for SLT. XmDA augments the source side by mixing sign embeddings with gloss embeddings, and it boosts the target side through knowledge distillation from gloss-to-text teacher models, without the need for additional data resources.

### 5.2 Mix-up Data Augmentation

Mix-up, an effective data augmentation technique, has made significant strides in the computer vision (CV) field (Zhang et al., 2017; Yoon et al., 2021; Liu et al., 2022; Hao et al., 2023; Zhang et al., 2023b). Its unique approach involves creating new training samples through linear interpolating a pair of randomly chosen examples and their respective labels at the surface level (Zhang et al., 2017) or feature level (Verma et al., 2019).

Recently, the natural language processing (NLP) domain has benefited from applying the mix-up technique. It has demonstrated promising outcomes in various tasks such as machine translation (Cheng et al., 2021), multilingual understanding (Cheng et al., 2022), as well as speech recognition and translation (Meng et al., 2021; Fang et al., 2022). As far as we know, XmDA is the first study to extend mix-up data augmentation to sign language translation tasks, encompassing visual and text modalities.

### 5.3 Knowledge Distillation

Knowledge Distillation (KD) (Hinton et al., 2015) is a powerful technique for model compression and transfer learning, wherein a smaller student model is trained to mimic the behavior of a larger, more complex teacher model or an ensemble of models, allowing the student to learn a more gen-

| System | Translation Output |
|--------|--------------------|
| Reference | **ich wünsche** ihnen noch einen schönen abend |
| Baseline | ihnen einen schönen abend und machen sie es gut |
| + XmDA | **ich wünsche** ihnen einen schönen abend |

Table 7: Comparison of translation outputs for low-frequency words generation. The word "ich wünsch" is emphasized to indicate its low-frequency nature in the dataset, i.e., frequency $\in (0, 10]$.

| System | Translation Output |
|--------|--------------------|
| Reference | da haben wir morgen schon die dreißig grad morgen im süden von frankreich auch und für uns wahrscheinlich schon im südwesten die fünfundzwanzig grad |
| Baseline | dort morgen bis dreißig grad im äußersten süden und auch im südwesten |
| + XmDA | da haben wir morgen auch die dreißig grad schon über frankreich und auch in süddeutschland haben wir noch die wärme schon mal über fünfundzwanzig grad im südwesten |

Table 8: Comparison of translation outputs for long sentence generation, i.e., length > 20. The sentences illustrate the model's capability to produce coherent and contextually appropriate translations for longer input sequences.

eralized and robust representation. This results in improved performance and efficiency, particularly in resource-constrained environments.

Within the NLP domain, KD has been successfully applied to a diverse range of tasks (Tang et al., 2019; Tan et al., 2019; Zhou et al., 2020; Jiao et al., 2020) and can be categorized into word-level (Wang et al., 2018) and sentence-level distillation (Kim and Rush, 2016), with the latter gaining prominence due to its superior performance and training efficiency.

In our work, we extend the concept of KD to cross-modality end-to-end SLT scenarios. In this setting, the end-to-end SLT student model learns to mimic the behavior of various gloss-to-text teacher models, leveraging the power of the teacher to improve end-to-end SLT performance.

## 6  Conclusion

In this paper, we propose a novel Cross-modality Data Augmentation (XmDA) approach to tackle the modality gap and data scarcity challenge in end-to-end sign language translation (SLT). XmDA integrates two techniques, namely Cross-modality Mix-up and Cross-modality Knowledge Distillation (KD). Cross-modality Mix-up bridges the modality gap by producing mixed-modal input representations, and Cross-modality KD augments the spoken output by transferring the generation knowledge from powerful gloss-to-text teacher models. With the XmDA, we achieve significant improvements on two widely used sign language translation datasets: PHOENIX-2014T and CSL-Daily.

Comprehensive analyses suggest that our approach improves the alignment between sign video features and gloss embeddings and the prediction of low-frequency words and the generation of long sentences, thus outperforming the compared methods in terms of BLEU, ROUGE and ChrF scores.

## 7  Limitations

We identify two limitations of our XmDA approach:

- The success of XmDA is contingent upon the quality of gloss-to-text teacher models, particularly those like PGEN (Ye et al., 2023). In environments with limited resources or unique data contexts, reproducing such models might be demanding, potentially affecting XmDA's generalization.

- The approach's efficacy is tied to the availability and integrity of gloss annotations. For sign languages or dialects lacking comprehensive gloss resources, the full potential of XmDA might not be realized, underscoring the value of rich gloss resources in sign language translation.

## 8  Acknowledgement

This research was supported in part by the National Natural Science Foundation of China (Grant No.92370204) and the Science and Technology Planning Project of Guangdong Province (Grant No. 2023A0505050111).

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

## A    Appendix

### A.1    Data Statistics

We evaluate the effectiveness of our proposed XmDA on two widely used sign language translation datasets: PHOENIX-2014T and CSL-Daily. These datasets provide diverse characteristics and contexts, making them ideal for our evaluation. Table 9 gives an overview of these datasets.

### A.2    Hyper-parameters of Baselines

Table 10 presents the hyper-parameters of Sign Language Transformers used in this work.

### A.3    The Optimal Components of XmDA

In this section, we present the ablation studies to explore the optimal components of our proposed XmDA approach. Unless otherwise stated, we primarily conduct the analyses on the German PHOENIX-2014T. Note that we strive to identify the near-optimal setting for our method mainly based on our experience rather than a full-space grid search, as aggressively optimizing the system requires significant computing resources and is beyond our means.

### A.3.1    What is the Optimal Mix-up strategy?

We investigate the optimal mix-up ratio $\lambda$ for our Cross-modality Mix-up. Inspired by Fang et al. (2022), we consider two types of strategies: a static strategy, where the $\lambda$ is fixed, and a dynamic strategy, where the $\lambda$ is adaptively determined based on the confidence $\mathcal{P}\left(\pi^* | h(\mathcal{S})\right)$ of the sign-gloss forced alignment (see Formula 4). Specifically, We constrain $\lambda$ in [0.0, 0.2, 0.4, 0.6, 0.8, 1.0] or $\lambda = Sigmoid(\mathcal{P}\left(\pi^* | h(\mathcal{S})\right) - 0.5)$ for experiments on PHOENIX-2014T. When $\lambda = 1.0$, all of the sign embeddings $\mathcal{F}$ will be swapped by gloss embedding $\mathcal{E}$ (e.g., $\mathcal{M} = \mathcal{F}$), while $\lambda = 0.0$ means none of the sign embeddings will be swapped into gloss embeddings and "+ Cross-modality Mix-up" degrades to the baseline model.

As shown in Figure 4, our observations reveal that: 1) for the static strategy (represented by the bleu line), Cross-modality Mix-up attains optimal SLT performance on the development set with mix-up ratio $\lambda = 0.6$; and 2) Cross-modality Mix-up employing the dynamic strategy (indicated by the red dotted line) marginally outperforms the static strategy with $\lambda = 0.6$. Considering dynamic strategy requires additional computation burden, we

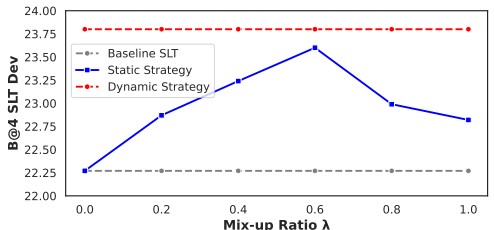

Figure 4: BLEU score of "+ Cross-modality Mix-up" on PHOENIX-2014T dev set, with different mix-up ratio $\lambda$. When $\lambda = 0.0$, "+ Cross-modality Mix-up" degrades to the baseline model.

choose the static strategy with $\lambda = 0.6$ as the Cross-modality Mix-up setting to balance performance and computational cost.

### A.3.2    What is the Optimal Number of Teacher Models?

As discussed in Section 2.4, we adopt the PGEN (Ye et al., 2023) to train multiple gloss-to-text teacher models, which will be used to obtain the new version of the target spoken language texts for SLT by translating the corresponding gloss annotations into spoken texts. We examine the optimal number of teacher models and the translation performance of the teacher models, sorted by their BLEU scores on the dev set, which can be found in Table 15.

As illustrated in Table 11, we examine the relationship between the number of teacher models and the performance of the end-to-end SLT model employing Cross-modality KD. The experimental results show that increasing the number of teacher models from K=0 to K=4 leads to a notable improvement in performance, while further increasing the number of teachers only provides marginal gains. To balance computational cost and model performance, we select K=4 as the default setting for the Cross-modality KD.

### A.4    Ablation Study of Components in XmDA

In this section, we evaluate the performance of the key components used in our Cross-modality Data Augmentation (XmDA) approach. These components, including the SMKD sign features extractor, the sign-gloss forced aligner, and the gloss-to-text teacher models, play crucial roles in the successful implementation of XmDA. Understanding their performance and effectiveness is essential for a comprehensive understanding of our proposed approach. The following subsections provide detailed

| Dataset | Language | #Signs | #Videos (avg.) | #OOV | #Signers | Topic | Source |
|---|---|---|---|---|---|---|---|
| | | | **Statitics** | | | | |
| PHOENIX-2014T (Camgoz et al., 2018) | DGS | 1,870 | 8,257 (9) | 41/1870 | 9 | Weather | TV |
| CSL-Daily (Zhou et al., 2021) | CSL | 2,000 | 20,654 (7) | 0/2,000 | 10 | Daily | Lab |

Table 9: Statistics of the two sign language benchmark datasets used in this work. DGS: German Sign Language; CSL: Chinese Sign Language; #Signs: number of unique glosses in the entire dataset; #Videos: number of sign videos in the entire dataset; avg.: average gloss in each video; #OOV: out-of-vocabulary glosses that occur in dev and test sets but not in the train set; #Signers: number of individuals in the entire dataset.

| Parameter | PHOENIX-2014T | CSL-Daily |
|---|---|---|
| encoder-layers | 3 | 1 |
| decoder-layers | 3 | 1 |
| attention heads | 8 | 8 |
| ctc-layers | 1 | 1 |
| hidden size | 512 | 512 |
| activation function | gelu | gelu |
| learning rate | $1 \cdot 10^{-3}$ | $1 \cdot 10^{-3}$ |
| Adam $\beta$ | (0.9, 0.98) | (0.9, 0.98) |
| label-smoothing | 0.1 | 0.1 |
| max output length | 30 | 50 |
| dropout | 0.3 | 0.3 |
| batch-size | 128 | 128 |

Table 10: Hyperparameters of Sign Language Transformer models.

| K | K=0 | K=1 | K=2 | K=4 | K=8 |
|---|---|---|---|---|---|
| BLEU | 22.90 | 23.54 | 24.33 | 24.82 | 24.93 |

Table 11: The impact of gloss-to-text teacher number. The BLEU of S2T references the BLUE-4 score on PHOENIX-2014T dev set with "+ Cross-modality KD".

evaluations for each of these components.

### A.4.1 Performance Evaluation of the SMKD Sign Features Extractor

Following Zhang et al. (2023a), we adopt the visual pre-trained model SMKD (Min et al., 2021) to extract sign video frame representations. Here, we report the Word Error Rate (WER) as the metric for sign features extractor, where measuring the similarity between the predicted gloss sequence and the ground truth. The results are listed in Table 12

### A.4.2 Evaluation of Sign-Gloss Forced Aligner Performance

In Section 2.3, as demonstrated, we perform Cross-modality Mix-up using the CTC classifier as a sign-

| Method | PHOENIX-2014T | | CSL-Daily | |
|---|---|---|---|---|
| | Dev | Test | Dev | Test |
| SMKD | 19.64 | 20.01 | 29.32 | 30.02 |

Table 12: Evaluation of the pre-trained SMKD sign features extractor on WER (%)(the lower the better).

gloss forced aligner to identify the temporal boundaries between glosses in continuous sign video frames. Considering that the CTC classifier is updated during the training stage, we report the performance of the aligner when the model training has reached a relatively stable state (e.g., epoch=25), thereby illustrating the effectiveness of the sign-gloss forced aligner. Specifically, we report the performance of the aligner for both the "+Cross-modality Mix-up" and the "+ XmDA" schemes in Section 3. As listed in Table 13, we present the WER on both the training and test sets to demonstrate the effectiveness and generalizability of the aligner.

| Method | PHOENIX-2014T | | CSL-Daily | |
|---|---|---|---|---|
| | Train | Test | Train | Test |
| Baseline SLT | 9.48 | 26.34 | 6.03 | 38.48 |
| + Mix-up | 9.93 | 26.01 | 5.86 | 37.60 |
| + XmDA | 7.66 | 25.84 | 4.65 | 33.41 |

Table 13: Evaluation of the CTC Classifier, i.e., sign-gloss aligner, effectiveness and generalizability with WER (%) (the lower the better).

### A.4.3 Performance Evaluation of the Gloss2Text teachers model

The Gloss2Text teacher models in our XmDA approach are crucial for Cross-modality Knowledge Distillation. Their performance directly impacts the quality of the generated spoken language texts used for softening target labels. We evaluate their translation performance in Table 15.

| Model | PHOENIX-2014T | | | CSL-Daily | | |
|---|---|---|---|---|---|---|
| | Precision | Recall | F1 | Precision | Recall | F1 |
| Sign Language Transformers | 0.8625 | 0.8884 | 0.8753 | 0.9037 | 0.9248 | 0.9142 |
| + Cross-modality Mix-up | 0.8685 | 0.8929 | 0.8805 | 0.9075 | 0.9275 | 0.9174 |
| + Cross-modality KD | 0.8691 | 0.8942 | 0.8815 | 0.9081 | 0.9272 | 0.9176 |
| + XmDA | **0.8945** | **0.9157** | **0.9051** | **0.9094** | **0.9288** | **0.9190** |

Table 14: BERTScore evaluation results for different models on PHOENIX-2014T and CSL-Daily Test datasets.

| ID | ID=0 | ID=1 | ID=2 | ID=3 | ID=4 |
|---|---|---|---|---|---|
| **BLEU** | 27.35 | 27.24 | 27.11 | 27.02 | 26.93 |
| **ID** | ID=5 | ID=6 | ID=7 | ID=8 | ID=9 |
| **BLEU** | 26.90 | 26.84 | 26.83 | 26.83 | 26.82 |

Table 15: Translation performance of the Gloss2Text teacher models on the dev set, sorted by BLEU score. ID refers to the unique identifier of the Gloss-to-Text teacher model

## A.5 BERTScore Evaluation

We have incorporated the BERTScore evaluation for a comprehensive assessment of our model's performance. This evaluation metric provides precision, recall, and F1 measures to quantify the quality of translations.

The results clearly indicate that our proposed XmDA model surpasses the baseline across all metrics, showcasing its superiority in terms of BERTscore.