# OpenReview forum: "Cross-modality Data Augmentation for End-to-End Sign Language Translation"
_EMNLP/2023/Conference — EMNLP 2023 Findings_

### Official Review · Reviewer_KVjs · 2023-08-02

**Soundness:** 4

**Excitement:**

3: Ambivalent: It has merits (e.g., it reports state-of-the-art results, the idea is nice), but there are key weaknesses (e.g., it describes incremental work), and it can significantly benefit from another round of revision. However, I won't object to accepting it if my co-reviewers champion it.

**Paper Topic And Main Contributions:**

This paper focus on end-to-end sign language translation, which aims at converting sign language videos into spoken language texts without generating itermediate gloss. To tackle the data scaricy, this paper utilizes  gloss-to-text model to perform multi-teacher sequence-level knowledge distillation . To tackle the modality gap, this paper leverages cross-modality mix-up to encourages the feature alignment. Experiments show the effectiveness of two introduced techniques and demonstrate the significant improvements over the SOTA.

**Reasons To Accept:**

1. This paper sucessfully introduces Cross-Modality Mixup and Cross-modality Knowledege Distillation, to tackle the modality gap and data scarity of end-to-end sign language translation. The experiments show that the proposed method significantly improves the SOTA (about 4BLEU on two dataset).

**Reasons To Reject:**

1. The two introduced techniques are well-studied in speech translation field. Leveraging mixup and JD loss to bridge the modality gap is proposed by [1]. Leveraging gloss-to-text model to translate gloss from video-gloss data into language text is very similar to leveraging MT model to translate the source text from ASR data into target text [2].

2. My main concern is the soundness of leveraging CTC as a sign-gloss forced aligner (Section 2.3). MFA (a HMM-GMM model) is widely used as speech-transcription forced aligner, there are two reasons why CTC classifier is discard.

   * Firstly, the CTC classifier is more likely to produce <blank> than repeat. Suppose the gloss is "i love cat", the CTC classifier is more likely to produce "_ i _ _ love _ _ cat _" than "i i love love love love cat cat cat". How to deal with the blanks to segment sign video?
   * Secondly, the CTC classifier may produce wrong output. Suppose the gloss is "i love cat", the CTC classifier may generate "i do like cat". How to deal with this session to perform force-aligned, especially the performance of CTC model is not good (according to table 8, the WER is about 20).

   If this paper makes extra efforts, they should describe in detail. Otherwise, the soundness of choosing CTC classifier as forced aligner should be discussed.

3. The analysis are not convincing.

   * To prove the effectiveness of mixup mechanism, there are only qualitative analysis without any quantitative analysis. Word-level similarity or sequence-level similarity are more convincing. Also, visulization of the representation produced by translation encoder will be more convincing than on the word embedding level.
   * In Table 3 and Table 4, this paper shows that the introduced two techniques work on different examples (based on word frequency and sequence length). The phenomenon is interesting, but there is no analysis.

[1] Fang Q, Ye R, Li L, et al. Stemm: Self-learning with speech-text manifold mixup for speech translation. ACL 2021.
[2] Jia Y, Johnson M, Macherey W, et al. Leveraging weakly supervised data to improve end-to-end speech-to-text translation. ICASSP 2019.

**Reproducibility:**

4: Could mostly reproduce the results, but there may be some variation because of sample variance or minor variations in their interpretation of the protocol or method.

**Reviewer Confidence:**

5: Positive that my evaluation is correct. I read the paper very carefully and I am very familiar with related work.

---

> ### Author Rebuttal · Authors · 2023-08-29
>
> We appreciate your time and effort in reviewing our paper. We summarize and address the comments below:
>
>     1. About the novelty
> Our XmDA method comprises two essential components: Xm Mix-up and XmKD. While the core concept of Xm Mix-up bears similarity to STEMM, as outlined in Section 4.1.1, we believe that our study's primary contribution lies in addressing data scarcity in sign language translation. This point will be further clarified in the revised version.
>
>     2. Soundness of leveraging CTC as a sign-gloss forced aligner (Section 2.3).
>
>  ● 1）The CTC classifier is a more general and prevalent component in sign language translation [1, 2, 3]. Our selection of the CTC classifier enables the sign-gloss forced aligner to function effectively within the end-to-end training setting.
>
> ● 2）Addressing concerns about the CTC alignment
>
> a. We do not need to address <blank> and incorrect output issues because the sign-gloss forced aligner segments sign video based on the predicted gloss distribution rather than the final CTC outputs.
>
> b. As shown at Eqt. (4), sign-gloss forced aligner is to search the best borders for certain ground truth glosses by maximizing the marginal probability. It would not consider the <blank>  and other tokens due to the constraints of ground truth glosses, e.g., "i love cat".
>
> ● 3） Regarding the performance of the CTC classifier, the cross-modal mixup is implemented during the training phase, yielding a Word Error Rate (WER) of 7.7,  as referenced in Table 9. This is on par with human performance[4].
>
> We appreciate your insightful comments and constructive recommendations. The suggested content will be incorporated into the revised version of the paper.
>
>     3. About the analysis.
>
> ● 1） Quantitative analysis using word-level similarity or sequence-level similarity
>
> a. Thank you for your suggestion. We have reproduced the visualization analysis based on the hidden representation of the final encoder layer. The outcomes align closely with our current findings. We will showcase these new results in our final version.
>
> b. In response to quantitative analysis, we utilized Kernel Density Estimation (KDE) to estimate the probability density functions on those tree types of embeddings[5,6]. The entropies resulting from these KDEs are as follows:
>
>
> | Embedding Type                  | KDEs Entropy |
> |---------------------------------|:------------:|
> | Gloss Embeddings                | 0.19         |
> | Sign Embeddings (Baseline)      | 2.18         |
> | Sign Embeddings (XmDA)          | 0.84         |
> Table 2: Entropies resulting from KDEs on different types of embeddings.
>
> c. Additionally, to provide a quantitative measure of alignment, we compared the average Euclidean distance and cosine similarity at the word level between sign embeddings and gloss embeddings. The results are detailed below:
>
>
>
>
> | Comparison                                      | Euclidean Distance | Cosine Similarity |
> |-------------------------------------------------|:------------------:|:-----------------:|
> | Sign Embedding (Baseline) vs. Gloss Embedding   | 14.6               | 0.19              |
> | Sign Embedding (XmDA) vs. Gloss Embedding       | 8.68               | 0.34              |
>
> Table 3: Average Euclidean distance and cosine similarity between sign embeddings and gloss embeddings.
>
> The above results show that our XmDA technique significantly improves the sign embedding representations over the baseline model.
>
> ● 2） Deeper analysis of the phenomenon in Table 3 and Table 4.
>
> Thank you for pointing out the phenomena observed in Tables 3 and 4. In response, we examined the outputs and identified intriguing trends through two case studies:
>
> **Case A: Low-frequency word prediction**
>
> | System       | Output                                                                        |
> |--------------|-------------------------------------------------------------------------------|
> | Ref          | **ich wünsche** ihnen noch einen schönen abend                                |
> | Baseline | ihnen einen schönen abend und machen sie es gut                               |
> | + Xm KD      | jetzt **wünsche ich** ihnen noch einen schönen abend und hoffe                |
> | + XmDA       | **ich wünsche** ihnen noch einen schönen abend                                |
>
> **Case B: Long sentence generation**
>
> | System          | Output                                                                                                                              |
> |-----------------|-------------------------------------------------------------------------------------------------------------------------------------|
> | Ref             | da haben wir morgen schon die dreißig grad morgen im süden von frankreich auch und für uns wahrscheinlich schon im südwesten die fünfundzwanzig grad |
> | Baseline    | dort morgen bis dreißig grad im äußersten süden und auch im südwesten                                                               |
> | + XmDA          | da haben wir morgen auch die dreißig grad schon über frankreich und auch in süddeutschland haben wir noch die wärme schon mal über fünfundzwanzig grad im südwesten |
>
>
> XmDA significantly improves translations for low-frequency words and long sentences over the baseline system. The case study will be incorporated in the revised version of the paper to illustrate this improvement further.
>
>
> **Reference**
>
> [1]  Sign Language Transformers: Joint End-to-end Sign Language Recognition and Translation CVPR 2020
>
> [2]  A Simple Multi-Modality Transfer Learning Baseline for Sign Language Translation CVPR 2022
>
> [3]  CVT-SLR: Contrastive Visual-Textual Transformation for Sign Language Recognition with Variational Alignment CVPR 2023
>
> [4]  Achieving human parity in conversational speech recognition  Microsoft Research Technical Report 2017
>
> [5]  Ensemble estimation of multivariate f-divergence IEEE 2014
>
> [6]  Density estimation for statistics and data analysis. CRC 1986

---

### Official Review · Reviewer_5AGz · 2023-08-03

**Soundness:** 4

**Excitement:**

3: Ambivalent: It has merits (e.g., it reports state-of-the-art results, the idea is nice), but there are key weaknesses (e.g., it describes incremental work), and it can significantly benefit from another round of revision. However, I won't object to accepting it if my co-reviewers champion it.

**Paper Topic And Main Contributions:**

This paper presents a training method that exploits the data triplet of "video (S), gloss-labels (G) and spoken-language-translation (T)" to reduce the modality gap in direct sign language translation (SLT). The two main components are cross-modality (Xm) 1) data augmentation (DA) and 2) knowledge distillation (KD).

"XmDA" extends the idea of Mixup used in images or speech to create pseudo-data without using extra data, such as self-training or back-translation. The visual input is transformed to a sequence of contextual representations by a pre-trained visual model "SMKD" whereas the corresponding "gloss" labels are transformed to a sequence of textual representations by a pre-trained "Gloss2Text" model. By using the visual-to-gloss alignment obtained from a CTC classifier, a pseudo sequence which contains (possibly) two modalities in its temporal dimension, depending on the sampled value and the mixing ratio, is generated for training. Its loss is a sum of Jensen-Shannon Divergence (JSD) (between the translation-label distribution of XmDA and that of visual input) over the time steps of the target.

"XmKD" uses multiple pre-trained "Gloss2Text" models to generate diverse targets (hard-labels), i.e, the spoken-language-translation for training.

Similar to previous work, it also uses a CTC loss to strength the alignment between "S" and "G" in the translation encoder and uses a cross-entropy loss to capture the alignment between "S" and "T" in the translation encoder-decoder. In total, there are 3 losses in training: 1) the CTC loss, 2) cross-entropy loss and the 3) JSD loss from XmDA.

Experimental results on 1) PHENIX-2014T dataset and 2) CSL-Daily dataset shows the effectiveness of "XmDA", "XmKD" and their combined effect. There are also studies on the improvement on low-frequency words and long inputs.

## Comment
The proposed method is sound and reasonable. The experimental results supports their claims. A drawback of the proposed method would be the reliance on the data triplets, e.g., the gloss-labels for modality mixing and CTC alignment. Since the data triplet is used, it would be informative to show the performance of a cascade system. Secondly, the novelty of the training method could be very weak for speech-translation researchers since it is almost identical to a series of work in reducing modality gap between speech inputs and textual inputs, in particular Fang et. al 2022 (STEMM: Self-learning with Speech-text Manifold Mixup for Speech Translation).




**Questions For The Authors:**

- Are the multiple references generated in "XmKD" also used in "MLE" (the cross-entropy between "S" and "T")? If yes, it is not clearly shown in Figure 1.

**Reasons To Accept:**

- The proposed method is sound and reasonable. The experimental results, together with the analysis, support the claims.
- The analysis on the prediction of low-frequency words and long inputs bring extra useful insights than just evaluation on metrics such as BLEU.

**Reasons To Reject:**

- The novelty of the training method could be very weak for speech-translation researchers since it is almost identical to a series of work in reducing modality gap between speech inputs and textual inputs, in particular Fang et. al 2022 (STEMM: Self-learning with Speech-text Manifold Mixup for Speech Translation).

**Reproducibility:**

4: Could mostly reproduce the results, but there may be some variation because of sample variance or minor variations in their interpretation of the protocol or method.

**Reviewer Confidence:**

3: Pretty sure, but there's a chance I missed something. Although I have a good feel for this area in general, I did not carefully check the paper's details, e.g., the math, experimental design, or novelty.

---

> ### Author Rebuttal · Authors · 2023-08-29
>
> We appreciate your time and effort in reviewing our paper. We summarize and address the comments below:
>
>     1. About the novelty
>
> Our XmDA method comprises two essential components: Xm Mix-up and XmKD. While the core concept of Xm Mix-up bears similarity to STEMM, as outlined in Section 4.1.1, we believe that our study's primary contribution lies in addressing data scarcity in sign language translation. This point will be further clarified in the revised version.
>
>     2. Multiple references used in XmKD
>
> Thank you for pointing that out. Indeed, we utilize the multiple references in the proposed XmKD. We apologize for any confusion caused by the oversight in Figure 1.  We will ensure that this is depicted in the revised version.

---

### Official Review · Reviewer_VzGU · 2023-08-04

**Soundness:** 4

**Excitement:**

4: Strong: This paper deepens the understanding of some phenomenon or lowers the barriers to an existing research direction.

**Paper Topic And Main Contributions:**

This paper is about end-to-end sign language translation. The authors have proposed an end-to-end system with several interesting components that obtain state of the art. Though the proposed work has some problems (see Reasons to reject), overall the ideas presented are model agnostic and thus can be applied across multimodal tasks, featuring its importance.

**Questions For The Authors:**

Is the cross-modality training taking place after training of the CTC classifier or is it end-to-end? It looks like there is a separate path for cross-modal mixup, and thus minimizing JSD loss between sign language distribution and multimodal distribution is not simultaneously taking place while also minimizing the MLE loss. Please explain them in section 2 clearly.

**Reasons To Accept:**

1. Well-motivated loss formulations.
2. Cross-modal knowledge distillation and transfer seem helpful for learning sign-gloss interaction, and in that sense, the paper is novel.
The authors demonstrate the superiority of their proposed method on two datasets: RWTH Phoenix 2014T and CSL-Daily.
3. Seem to work better for Long sentences.

**Reasons To Reject:**

1. Authors are suggested to use other metrics to evaluate the Results (e.g. BERTScore).
2. Often it is not sufficient to show automatic evaluation results. The author does not show any human evaluation results and does not even perform a case study and proper error analysis. This does not reflect well on the qualitative aspects of the proposed model.
3. It is difficult to understand the methodology without Figure 1. Parts of section 2 should be written in alignment with Figure 1, and the authors are expected to follow a step-by-step description of the proposed method. (See questions to authors)

**Reproducibility:**

3: Could reproduce the results with some difficulty. The settings of parameters are underspecified or subjectively determined; the training/evaluation data are not widely available.

**Reviewer Confidence:**

4: Quite sure. I tried to check the important points carefully. It's unlikely, though conceivable, that I missed something that should affect my ratings.

---

> ### Author Rebuttal · Authors · 2023-08-29
>
> We appreciate your time and effort in reviewing our paper.   We summarize and address the comments below:
>
>     1. Other metrics
> Following your recommendation, we have incorporated the BERTScore evaluation. The results, which include precision, recall, and F1 measures, are presented below:
>
>
> | Model                           |     |     PHOENIX-2014T    |       |         |   CSL-Daily      |       |
> |-------------------------------|:-------------------:|:-----:|:-----:|:------------------:|:-----:|:-----:|
> |                                 | Precision          | Recall| F1    | Precision          | Recall| F1    |
> | Sign Language Transformers      | 0.8625             | 0.8884| 0.8753| 0.9037             | 0.9248| 0.9142|
> | + Cross-modality Mix-up         | 0.8685             | 0.8929| 0.8805| 0.9075             | 0.9275| 0.9174|
> | + Cross-modality KD             | 0.8691             | 0.8942| 0.8815| 0.9081             | 0.9272| 0.9176|
> | + XmDA                          | **0.8945**         | **0.9157**| **0.9051**| **0.9094**     | **0.9288**| **0.9190**|
>
> Table 1: BERTScore evaluation results on PHOENIX-2014T and CSL-Daily Test datasets.
>
> The results show that the proposed XmDA outperforms the baseline in terms of BERT-score.
>
>     2. Case study and proper error analysis
>
> To delve deeper into our model's performance, we examined the outputs and identified intriguing trends.
>
> ● Low-frequency word prediction
> | System |  Output |
> | ------ | ------ |
> | Reference | **ich wünsche** ihnen noch einen schönen abend |
> | Baseline | ihnen einen schönen abend und machen sie es gut |
> | + XmDA |  **ich wünsche** ihnen einen schönen abend |
>
> ● Long sentence generation
> | System |  Output |
> | ------ | ------ |
> | Reference | da haben wir morgen schon die dreißig grad morgen im süden von frankreich auch und für uns wahrscheinlich schon im südwesten die fünfundzwanzig grad |
> | Baseline | dort morgen bis dreißig grad im äußersten süden und auch im südwesten |
> | + XmDA |  da haben wir morgen auch die dreißig grad schon über frankreich und auch in süddeutschland haben wir noch die wärme schon mal über fünfundzwanzig grad im südwesten |
>
> XmDA significantly improves translations for low-frequency words and long sentences over the baseline system. The case study will be incorporated in the revised version of the paper to illustrate this improvement further.
>
>      3. About framework description (Figure 2 and Section 2)
>
> We appreciate the feedback regarding Figure 1 and Section 2 of our paper. The proposed XmDA represents a joint training framework. In Figure 2, both dotted and solid lines operate simultaneously to train the model, with the dotted line specifically illustrating the forward path for mixed-up samples. This will be further clarified in the revised version.

---

### Meta-Review · Area_Chair_NQk4 · 2023-09-15

**Recommendation:** 3

**Metareview:**

The paper proposes cross-modality data augmentation techniques to tackle the data scarcity issues while training end-to-end sign language translation models. In particular, the paper proposes to use cross-modal mix-up and cross-modality knowledge distillation. The experimental results demonstrate the superiority of the proposed method.

The reviewers liked the method description and analysis. They asked several qualitative and quantitative evaluation questions, but the authors clarified them during the rebuttal. Finally, reviewers pointed out that while the techniques are novel in sign language translation, they are well-studied in speech translation, limiting their excitement.

---

### Decision · Program_Chairs · 2023-10-07

**Decision:**

Accept-Findings

**Comment:**

The paper proposes cross-modality data augmentation techniques to tackle the data scarcity issues while training end-to-end sign language translation models. In particular, the paper proposes to use cross-modal mix-up and cross-modality knowledge distillation. The experimental results demonstrate the superiority of the proposed method.

The reviewers liked the method description and analysis. They asked several qualitative and quantitative evaluation questions, but the authors clarified them during the rebuttal. Finally, reviewers pointed out that while the techniques are novel in sign language translation, they are well-studied in speech translation, limiting their excitement.